

# Contribution of temporal data to predictive performance in 30-day readmission of morbidly obese patients

Petra Povalej Brzan[1,2], Zoran Obradovic[3] and Gregor Stiglic[1,2]

[1] Faculty of Health Sciences, University of Maribor, Maribor, Slovenia
[2] Faculty of Electrical Engineering and Computer Science, University of Maribor, Maribor, Slovenia
[3] Center for Data Analytics and Biomedical Informatics, Temple University, Philadelphia, PA, United States

## ABSTRACT

**Background.** Reduction of readmissions after discharge represents an important challenge for many hospitals and has attracted the interest of many researchers in the past few years. Most of the studies in this field focus on building cross-sectional predictive models that aim to predict the occurrence of readmission within 30-days based on information from the current hospitalization. The aim of this study is demonstration of predictive performance gain obtained by inclusion of information from historical hospitalization records among morbidly obese patients.

**Methods.** The California Statewide inpatient database was used to build regularized logistic regression models for prediction of readmission in morbidly obese patients ($n = 18,881$). Temporal features were extracted from historical patient hospitalization records in a one-year timeframe. Five different datasets of patients were prepared based on the number of available hospitalizations per patient. Sample size of the five datasets ranged from 4,787 patients with more than five hospitalizations to 20,521 patients with at least two hospitalization records in one year. A 10-fold cross validation was repeted 100 times to assess the variability of the results. Additionally, random forest and extreme gradient boosting were used to confirm the results.

**Results.** Area under the ROC curve increased significantly when including information from up to three historical records on all datasets. The inclusion of more than three historical records was not efficient. Similar results can be observed for Brier score and PPV value. The number of selected predictors corresponded to the complexity of the dataset ranging from an average of 29.50 selected features on the smallest dataset to 184.96 on the largest dataset based on 100 repetitions of 10-fold cross-validation.

**Discussion.** The results show positive influence of adding information from historical hospitalization records on predictive performance using all predictive modeling techniques used in this study. We can conclude that it is advantageous to build separate readmission prediction models in subgroups of patients with more hospital admissions by aggregating information from up to three previous hospitalizations.

Corresponding author
Gregor Stiglic, gregor.stiglic@um.si

## INTRODUCTION

Hospital readmission prediction models have been widely studied and deployed worldwide (*Zhu et al., 2015*; *Hao et al., 2015*; *Stiglic et al., 2015*). Different types of prediction models were proposed to predict and potentially prevent hospital readmissions. As described in a review study by *Kansagara et al. (2011)*, we can divide the proposed predictive models into two groups—i.e., models relying on retrospective administrative data and models using real-time administrative data. While the second group usually focuses on data that is collected during hospitalization, the first group of models relies on retrospective data. Although many studies include information on prior hospitalizations to improve the predictive performance of readmission prediction models, they usually do not provide evidence on the level of their contribution to predictive performance.

*He et al. (2014)* demonstrated the importance of a logistic regression predictor representing the number of prior hospitalizations in the past five years. This simple variable was selected as significant in both a general and a specific chronic pancreatitis subgroup based predictive model. *Walsh & Hripcsak (2014)* compared predictive performance of readmission prediction models for specific subgroups of patients. Their results show a strong gain in predictive performance when laboratory data and visit history data is included in the predictive model development. A study by *Shahn, Ryan & Madigan (2015)* incorporates relative temporal relationships among multiple health events in the space of predictors to build a Random Relational Forest (RRF) based classifier originally developed in the context of speech recognition. RRF generates informative labeled graphs representing temporal relations among health events at the nodes of randomized decision trees. Although the target of a study by Shahn et al. does not include readmission classification, but focuses on predicting strokes in patients with prior diagnoses of Atrial Fibrillation, it demonstrates the importance of temporal information to achieve meaningful improvements in predictive performance. Similarily, the use of temporal information from electronic health records (EHRs) for prediction of Anastomosis Leakage was demonstrated in a study by *Soguero-Ruiz et al. (2016)*. The predictive performance gain was proven when combining the data from heterogenus data sources (extracted free text, blood samples values, and patient vital signs).

Morbidly obese patients represent one of the most complex populations in healthcare systems and are often related to higher treatment costs (*Kadry et al., 2014*). As outlined by *Incavo & Derasari (2014)*, hospitals need extra personnel and equipment to lift and transport morbidly obese patients. Additionally, such patients tend to have above average number of comorbidities, including chronic diseases. Consequently, hospital staff is affected from the heavy lifting of patients and healthcare providers need to provide more care with the same reimbursement (*Choi & Brings, 2016*).

In this study, we focus on the following research question: Does the inclusion of additional information from historical patient hospitalization records improve the predictive performance of 30-day readmission models? Hospitalization claims data from morbidly obese patients were used to build a readmission prediction model. We hypothesized that a significant improvement in predictive performance can be obtained by inclusion of new predictors based on previous hospitalizations. Least Absolute Selection
and Shrinkage Operator (Lasso) regularization (*Tibshirani, 2011*) was used to allow interpretation of the results by observing the frequency of the predictor inclusion (*Stiglic, Davey & Obradovic, 2013*) in the Lasso logistic regression model. We further explored the complexity of built models measured as number of selected features in addition to comparing the results using advanced predictive modeling techniques.

## MATERIALS & METHODS

### Dataset

The 11,889,326 hospitalization records from the Healthcare Cost and Utilization Project (HCUP) State Inpatient Database for California (SID CA) for the years 2009–2011 (*HCUP State Inpatient Databases (SID), 2009–2011*) were used in the study. Each hospitalization record includes demographic information about the patient (age, birth year, sex, race, etc.), information about the hospital stay (length of stay, total charges, type of payment, discharge month, survival information, scheduled visit, etc.), primary diagnosis, up to 25 diagnoses and up to 25 procedures performed on a patient during hospitalization. For the purpose of protecting patient privacy the hospitalization records are anonymized and a unique ID value, which enables tracking the patients through several years, is used for each patient. The diagnoses and procedures are described in ICD-9 codes and CSS codes. In our experiment the ICD-9 codes were used.

The HCUP SID CA dataset was filtered based on the following inclusion criteria

The initial database was first filtered on hospitalization records with valid patient ID on 8,373,831 hospitalization records from 4,674,262 patients. In the next step 237,773 patients (640,883 hospitalization records) with ICD-9 code 278.01 (morbid obesity) in at least one hospitalization were extracted from the database. The most recent hospitalization with ICD-9 code 278.01 from 2010 to 2011 was selected for each patient. Additionally, all historical records in a one year timeframe were added for each patient. In further experiments we position the patient in one hospitalization before the last one (index hospitalization) with the purpose of predicting the last hospitalization. All scheduled predicted admissions were excluded. For that purpose, five different datasets of patients were prepared based on the number of available hospitalizations per patient. In further text let $D_1$ denote the dataset of 20,521 patients with at least two hospitalization records in one year timeframe. Similarly, $D_2$ consists of 18,881 patients with more than two hospitalizations, $D_3$ dataset represents 11,603 patients with more than three hospitalizations and $D_4$ 7,413 patients with more than four hospitalization records. The smallest dataset $D_5$ includes 4,787 patients with more than five hospitalization records in a one year timeframe (Fig. 1).

The percentage of patients readmitted in 30 days increases from 34.36% in the largest dataset $D_1$ to 43.85% in the smallest dataset of the most complex patients ($D_5$). Note that the purpose of this study is to evaluate the contribution of additional information from historical records on the prediction of the 30-days readmission of morbid obesity patients. Therefore we used the patients that were readmitted at least once, because these patients represent a cohort of patients that are costly and have a strong impact on hospital and global performance indicators such as waiting lists, mortality, planned care, etc.

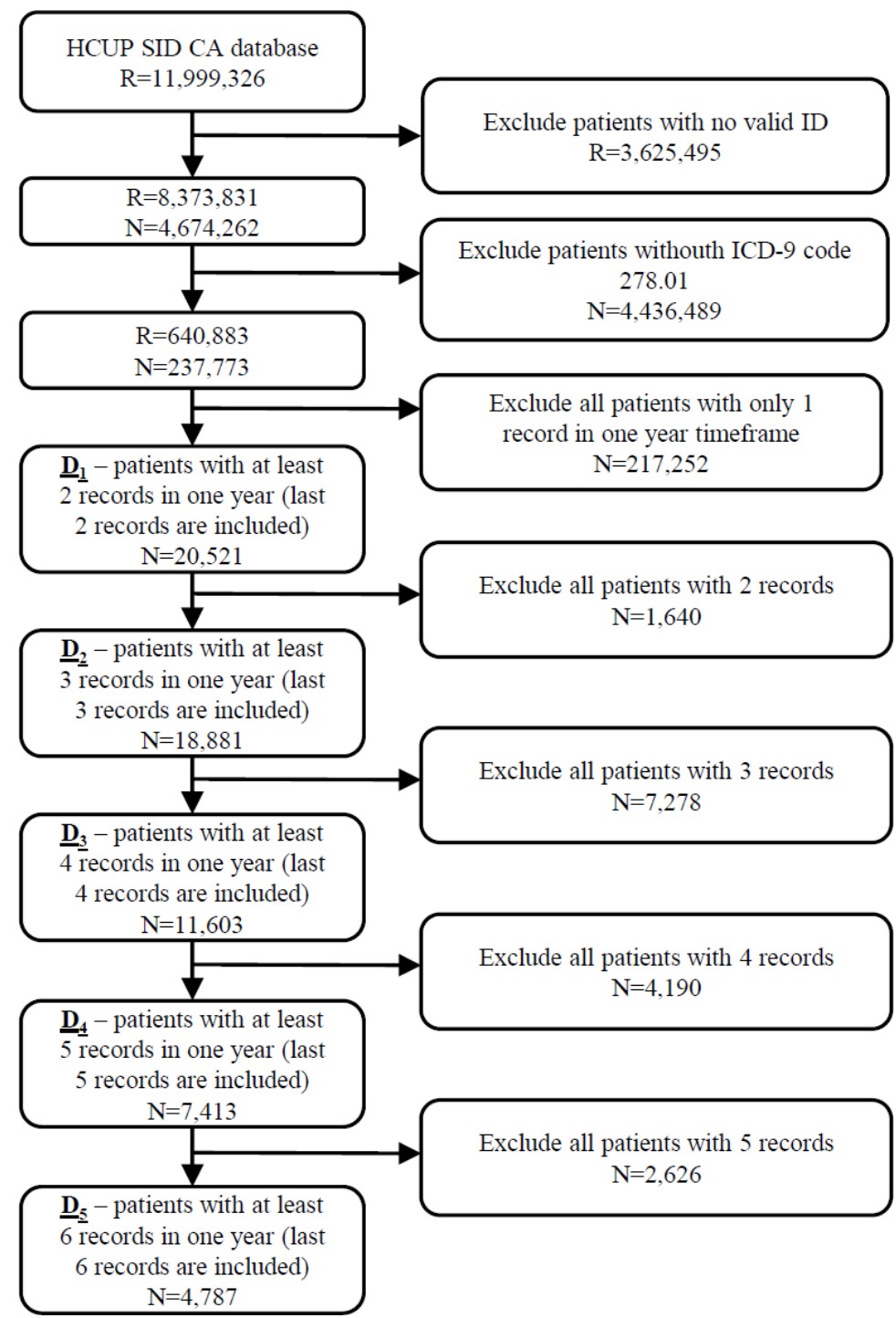

**Figure 1** **Extraction of datasets from the original HCUP SID CA database including number of records (R) and patients (N).**

All available diagnoses (7,196) and procedures (2,488) in the year 2009 were first ordered by frequency to arbitrarily select a cut-off value for selection of diagnoses and procedures used in later stages. Due to the long tail distribution of frequencies the cut-off was set to 3%, resulting in the final set of 217 diagnoses and 75 procedures that were further used as dichotomous variables.

## Statistical analysis

The following features from consecutive hospital records were obtained for each patient on the discharge day: total number of hospital days in all hospitalizations, total number of hospitalizations, total number of procedures in all hospitalizations, total number of chronic diseases in all hospitalizations, mean number of chronic diseases from all hospitalizations, mean number of hospital days from all hospitalizations, mean number of procedures from all hospitalizations. Additionally, a total number of hospital days, total number of hospitalizations, mean number of chronic diseases, mean number of hospital days, and mean number of procedures were calculated for the last 30/60/90/180/270 days prior to index hospitalization.

The number of occurrences of each diagnosis and procedures from all historical patient records were added as new features for each patient hospitalization record. The patient hospitalization record used in futher analysis therefore consisted of patient's demographic information (age, birth year, sex, race, etc.) and a set of features from historical hospital records obtained as described above.

### *Predicitve modeling*

A generalized linear model via penalized maximum likelihood combining L1-norm (lasso) and L2-norm (ridge) regularization was used as defined by *Friedman, Hastie & Tibshirani (2013)*. The process of predictive modeling can be significantly simplified by using regularized logistic regression methods, since the feature selection step is integrated in the process.

A generalized linear model via penalized maximum likelihood can be described as:

$$\min_{\beta_0, \beta} \frac{1}{N} \sum_{i=1}^{N} w_i l(y_i \beta_0 + \beta^T x_i) + \lambda \left[ (1 - \alpha) \|\beta\|_2^2 / 2 + \alpha \|\beta\|_1 \right]$$

where $i$ represents observations and it's negative log-likelihood contribution is noted as $l(y, \eta)$.

The regularization path $\lambda$ is computed for a grid of values for the regularization parameter which controls the overall strength of the penalty and is in our case calculated for lasso ($\alpha = 1$), since our initial experiments did not show any significant gain with elastic-net.

Additionally, the results of the lasso predictive model were compared to random forest (using 200 decision trees) and XGBoost method using default parameter values and the same experimental settings. Random forest is often applied in healthcare prediction problems since it offers a high level of robustness (*Zhou et al., 2016*). XGBoost is a powerful implementation of gradient boosting first proposed by *Friedman & Jerome (2001)* designed for speed and performance.

*Experimental Setting*

Each experimental run included 10-fold cross-validation repeated 100 times with the same random number generator seed values in all experiments. By using 100 repetitions of 10-fold cross validation, we were able to assess the variance of the results under different cross-validations.

The following experimental setup was used:

1. Select the dataset of patients with at least R hospitalization records in a one year timeframe.
2. Predict 30-days readmission for the last hospitalization record from the temporal features for 1, 2,..., R historical records, using 10-fold cross-validation.
3. Repeat step 2 for 100 times.
4. Repeat steps 1–3 for $R = \{1, 2, 3, 4, 5\}$.

Predictions were obtained for each validation sample using the model derived on the derivation sample. The predictive accuracy of each model was summarized by area under the receiver operating characteristics (ROC) curve (AUC), where 1 represents perfect predictive performance and 0.5 represents random performance. In addition to AUC, Brier's score, which allows more efficient evaluation of probabilistic predictions, was used. In contrast to AUC, lower Brier score represents better predictive performance. Due to imbalanced datasets, sensitivity, specificity, positive predictive value (PPV) and negative predictive value (NPV) were calculated. All evaluation measures were calculated for each 10-fold run on validation samples and then the mean value was calculated. The average of 10-fold mean over 100 repetitions and 95% confidence interval for each experiment is presented in all figures.

The independent samples t test was used to test the difference in mean evaluation measures between two samples. The ANOVA test was used for comparing the mean values for different datasets. $P$ value less than 0.05 was considered statistically significant.

All experiments were conducted using the R (*R Core Team, 2016*).

# RESULTS

The results in the text are presented as mean AUC (95% CI). The detailed results are described in Supplemental Information 1. Number of selected features and AUC on different datasets are presented in Fig. 2. The AUC on the smallest dataset ($D_5$) of 4,787 patients with at least five historical hospitalizations in a one-year timeframe increased from 0.631 (0.627, 0.636), when only current hospitalization was considered, to 0.657 (0.652, 0.0.661) when temporal features from the last 2 hospitalizations were included. Information from an additional (third) historical hospitalization again significantly improves AUC, which increases to 0.670 (0.665, 0.674). However, when the fourth and fifth historical hospitalization is added, the AUC increases only for 0.04 in each case (AUC = 0.674 (0.669,0.679) for four hospitalizations and AUC = 0.678 (0.673,0.683) for five hospitalizations). Similar trends can be seen when we reduce the required minimum number of historical hospitalization records in a one-year timeframe to four, three and two records ($D_4$, $D_3$, $D_2$) and consequently increase the sample size. The most significant
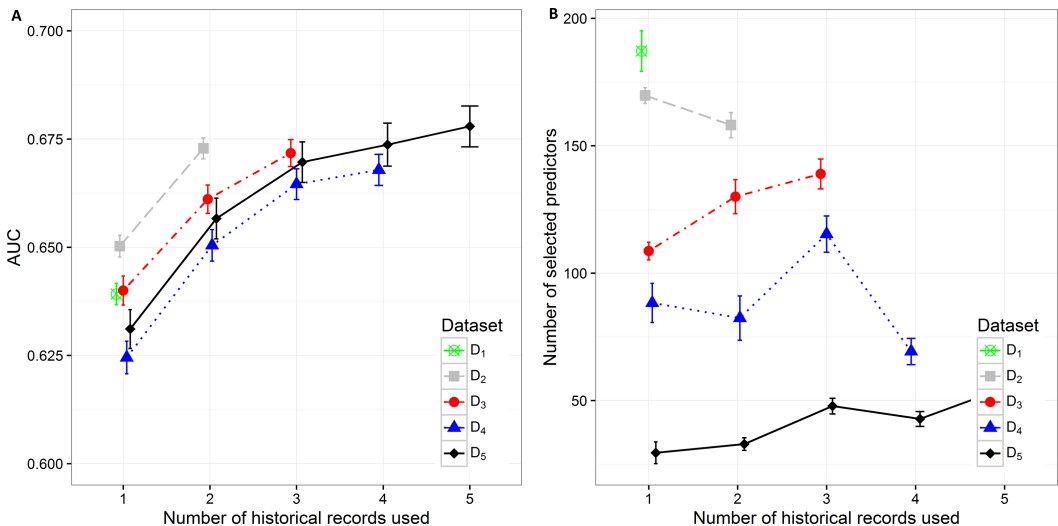

**Figure 2** Mean value of the (A) AUC and (B) number of selected predictors with corresponding 95% CI on different datasets with different number of historical hospitalization records.

difference in mean AUC can be observed when at least one historical hospitalization record is included. The AUC increases from 0.650 (0.648, 0.653) to 0.673 (0.670, 0.675) on the largest dataset $D_2$. Similar observations can be made on $D_3$, $D_4$, and $D_5$. As seen on the smallest dataset $D_5$, the statistical difference in AUC between the classifiers with two hospitalization records and three hospitalization records considered can be observed on bigger datasets ($D_4$ and $D_3$) as well. For databases $D_4$ and $D_5$ the experiment was repeated by adding a fourth hospitalization record. The increase in AUC was only 0.03 for $D_4$ (from 0.665 (0.661,0.668) to 0.668 (0.664,0.671)) and 0.04 for $D_5$ (from 0.670 (0.665,0.674) to 0.678 (0.673,0.683)).

By increasing the dataset size, the percentage of positive samples (patients that were re-hospitalized in 30 days) is decreasing. This is the most plausible reason for higher initial AUC when only current hospitalization is considered for prediction of re-hospitalization on bigger datasets. The highest mean value of AUC (0.650 (0.648, 0.653)) can be observed on the dataset $D_2$ with 18,881 patients and the lowest AUC (0.625 (0.621, 0.628)) on the dataset $D_4$ with 7,413 patients. When adding one more historical hospitalization record, the mean AUC value ranges between 0.650 (0.647, 0654) on $D_4$ and 0.673 (0.670, 0.675) on $D_2$. The difference is significant between all datasets except between $D_3$ (AUC = 0.661 (0.658, 0.664)) and $D_5$ (AUC = 0.657 (0.652, 0.661)) and between $D_4$ (AUC = 0.650 (0.647, 0.654)) and $D_5$ (AUC = 0.657 (0.652, 0.661)). However, when the third historical record is added, the mean AUC value ranges between 0.665 (0.661, 0.668) on $D_4$ and 0.672 (0.669, 0.675) on $D_3$. The difference in mean AUC is significant between $D_3$ and $D_4$. On the other hand, adding additional historical records (more than 3) does not result in significantly better AUC.

By increasing the size of the dataset we can observe the decreasing trend of the Brier score (mean squared error), which can be expected considering that with the size of the dataset the percentage of positive samples decreases (Fig. 3). When observing separate

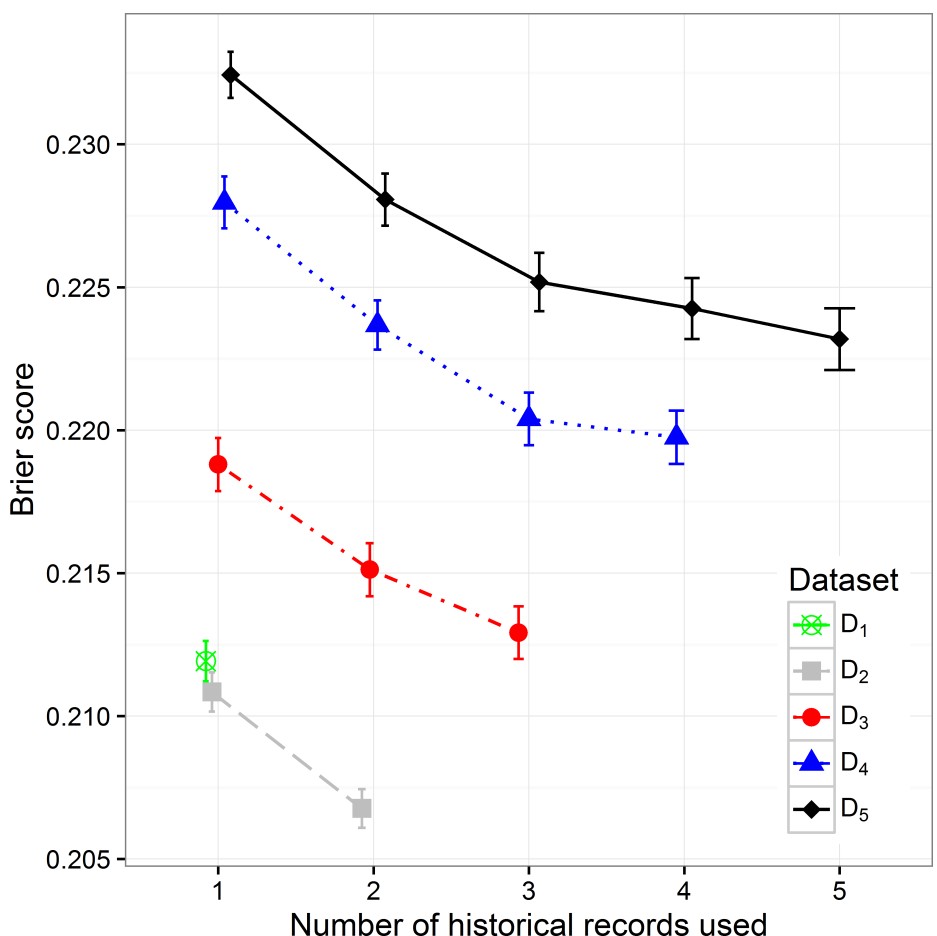

**Figure 3** The Brier score mean value and 95% CI on different datasets with different number of historical hospitalization records.

datasets, we can also see the decrease in Brier score when including additional historical hospitalization records. The decrease is statistically significant in models that include from one to three historical records. The inclusion of more historical records does not further decrease the score in a significant way.

The sensitivity value and PPV value trends can be observed in Figs. 4A and 4B. The lowest sensitivity (0.570 (0.564, 0.576)) and PPV value (0.448 (0.445, 0.452)) are shown on the biggest dataset $D_1$. The PPV value increases significantly when decreasing the size of the dataset. The highest PPV value (0.544 (0.538, 0.551)) when only current hospitalization data are included is therefore achieved on the smallest datast $D_5$ The sensitivity and PPV increase also when additional information about historical hospitalizations is included. The highest difference can be observed when at least one additional hospitalization record is added to the current hospitalization data. In all datasets this difference is significant. When adding the third hospitalization record the mean PPV value statistically still increases on all datasets, however the mean sensitivity value does not change significantly. The highest mean PPV value of 0.578 (0.572, 0.584) was achieved in the smallest dataset ($D_5$) when

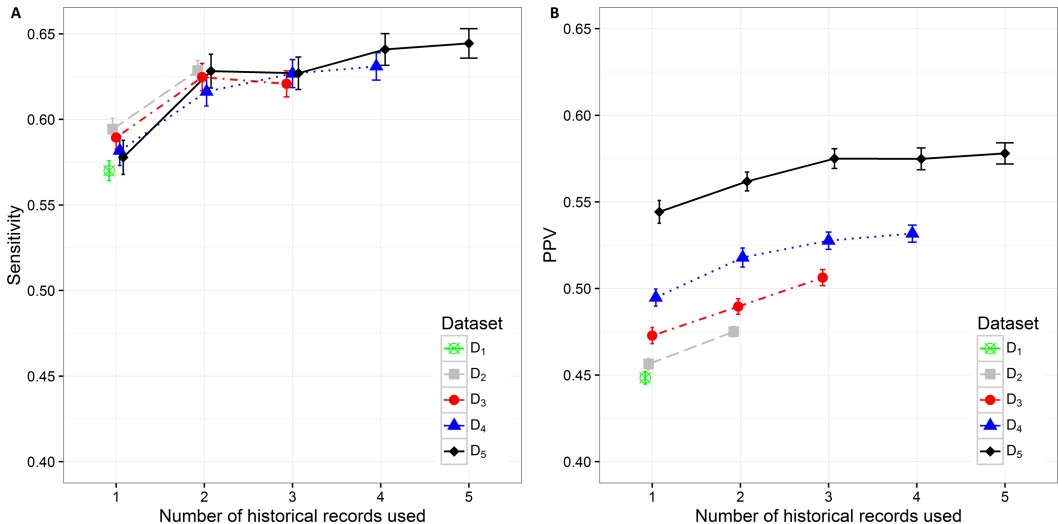

**Figure 4** Mean value of the (A) sensitivity and (B) PPV with corresponding 95% CI on different datasets with different number of historical hospitalization records.

the data from all five historical hospitalization records was included. However, it does not change significantly from the PPV value for $D_5$ when only three historical records are considered (0.575 (0.569, 0.581)).

The sensitivity changes significantly only when adding one historical record to the current hospitalization data on all datasets. The differences vary from 0.034 on $D_2$ to 0.050 on $D_5$. The highest sensitivity value of 0.644 (0.636, 0.653) was achieved on the smallest dataset $D_5$ when using the data from all five hospitalizations. However, the observation has to be made that it is not significantly different from the sensitivity value for $D_5$ when only two historical records are considered (0.628 (0.618, 0.638)).

Specificity varies from minimum 0.601 (0.593, 0.610) on $D_4$ with only current hospitalization record considered to maximum 0.643 (0.636, 0.650) on $D_3$ with historical information on all three records included. The increase in specificity while adding new historical information can be observed on all datasets except in $D_5$. As already observed in other measures, the statistically significant increase is shown between one and three hospitalization records included on all datasets. A very similar trend can be observed for NPV values, which also increase when adding historical information from previous hospitalizations (Fig. 5).

The number of selected predictors was considered as a measure of classifier complexity, where lower complexity is considered as positive when interpretation of results from the medical point of view is needed. Figure 2B shows that the lowest complexity in the terms of the mean number of selected predictors can be observed on the smallest dataset ($D_5$) where mean value increases from 29.50 (25.264, 33.776) to 54.98 (51.864, 58.096) with addition of previous hospitalization information. A similarly increasing trend can be observed on $D_3$ (from 108.700 (105.242, 112.158) to 138.950 (133.100, 144.800) features). The decrease in the number of selected predictors when adding historical records is shown on $D_2$ (from 169.7 (166.629, 172.811) to 158.130 (153.233, 163.027)) and partly on $D_4$. On $D_4$ with

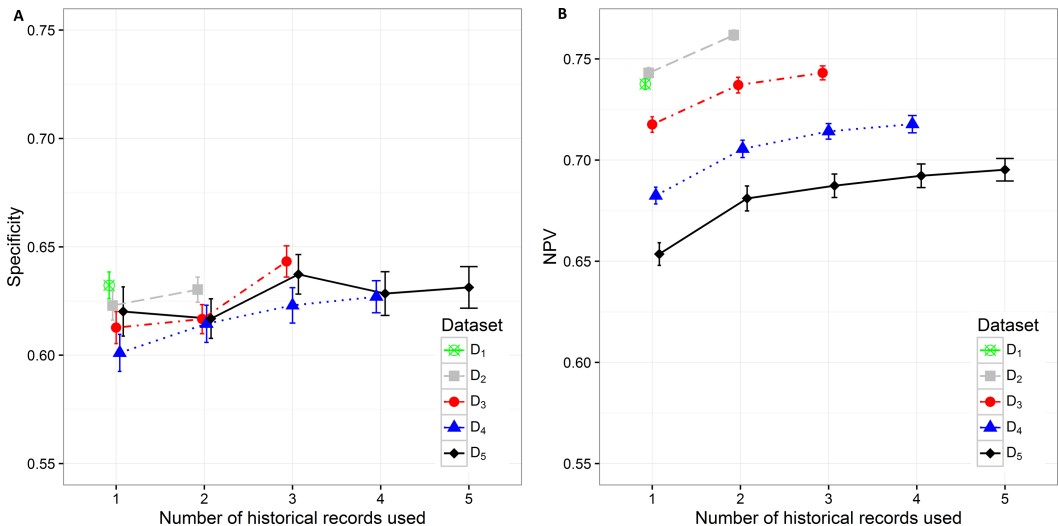

**Figure 5** Mean value of the (A) specificity and (B) NPV with corresponding 95% CI on different datasets with different number of historical hospitalization records.

three historical records included the mean number of selected predictors increases from 82.390 (73.700, 91.080) (for two historical records) to 115.330 (108.193, 122.467) and then decreases again to 69.28 (64.120, 74.440) when all four historical records are included. The highest mean number of selected features was obtained from the largest dataset $D_1$ (184.960 (179.257, 195.083)).

Additional experiments using random forest and XGBoost were performed. The results (Supplemental Information 2, 3) show similar trends as can be seen in the figures above (Figs. 2– 5). Detailed results including all performance metrics with corresponding confidence intervals for each predictive modeling technique can be found in Supplemental Information 1.

## DISCUSSION

Although claims data are very limited in information regarding a specific patient, claims databases usually contain large volumes of data and are accessible to researchers. In this study, we demonstrated how information on historical hospitalizations influences the predictive performance of a classifier built with a Lasso regularized generalized linear model on morbidly obese patients.

The initial idea was to show that more complex patients, which are more frequently hospitalized, are more similar to each other and therefore separate prediction models should be used for them. In order to make a fair comparison, five datasets were constructed from the initial database based on the minimum number of historical hospitalizations per patient ($D_1$–$D_5$). Then the model for 30-days readmission risk prediction was built using the data from one or more historical hospitalization records for each patient in the dataset. Each model was evaluated using a validation sample. The robustness of the models was tested using 10-fold CV, which was repeated 100 times.

The complexity of models was expressed by the number of selected predictors. Generally, higher mean AUC, sensitivity, specificity, PPV and NPV was achieved when more historical information was added. However, the initial models using only the data from the current hospitalization performed better on less restrictive datasets.

It is rather difficult to provide a guideline as to how many historical records should be included in a predictive model. We expect the optimal model to have an agreement between the performance measures (for example high AUC and low Brier score). However, the complexity of the model should also be considered. The simplest model should be selected for practical purposes of model interpretability.

The presented analysis of the influence of adding historical data on the predictive performance of a classifier provided very useful insights. When considering all performance measures and also the complexity of the classifiers, one can observe that it is highly important to build the prediction model using at least information from the last two hospitalization records if available. Including data from more than three historical records did not improve the performance of classifiers significantly. Therefore we can conclude that the inclusion of data for more than three previous hospitalization records per patient is not required. Results obtained with random forest and XGBoost predictive models (Supplemental Information 1) confirm the same trends that can be seen in all performance metrics for lasso logistic regression (Figs. 2– 5).

The analysis of model complexity on the basis of the number of selected predictors shows that building separate models for patients with a higher number of hospitalizations is reasonable, since the models built on more restrictive (homogeneous) datasets gain on simplicity and predictive performance. On the other hand, one can expect higher stability of prediction models on larger sample size (Figs. 2A, 4 and 5). Although we obtained the best results, measured by AUC, using random forest, it should be noted that these models used significantly more features in comparison to the other two techniques. XGBoost achieved the best Brier score performance. However, as in random forest the interpretability of XGBoost models is very limited compared to lasso logistic regression.

Readmissions represent one of the most important indicators of quality of care in the healthcare environment, resulting in great economic impact. Readmission rates within 30 days are reported as high as 19.6%, including approximately 76% of preventable readmissions, resulting in a reduction of about $25 billion annually in the US (*Behara et al., 2013*). Therefore, a robust and efficient solution to predict readmissions contributes to higher quality of care and reduces costs. Moreover, this work focuses on a specific subgroup of morbidly obese patients where nurses and nursing assistants manually lifting patients experience the highest rates of back and shoulder musculoskeletal injuries (*Choi & Brings, 2016*), such that an effective predictive model multiplies the benefits on both the patient and hospital staff side.

The limitations of this study are related to the characteristics of the claims data, where only a limited set of features is available. Additionally, it is important to realize that some diagnoses are not correctly recorded due to the influence of health insurance policies on costs related to different diagnoses and procedures. In the case of more complete data (full EHR records) the general predictive performance is expected to be higher; however,

the contribution of the historical data would still be significant. On the other hand, the availability and volume of claims data were more important in order to achieve the aim of this study.

## CONCLUSIONS

Existing literature on readmission prediction based on claims data shows relatively low predictive performance. Therefore, this study does not only focus on improvement of the predictive performance but also includes the analysis of how historical information about the patient influences the predictive performance and complexity of the predictive model.

The presented results show positive influence, which reflects in statistically significant increase in AUC, sensitivity, specificity, PPV and NPV values and decrease in Brier score when adding up to three historical hospitalization records.

As expected, the number of selected predictors increases with the size of the dataset. The homogeneity of the patient's dataset increases when tightening the criteria regarding minimum number of hospitalizations per patient in a one year timeframe.

This study can be usefull for data-scientists and software engineers developing similar prediction models using data from hospitalization records. Most already developed readmission prediction models are focused on readmission prediction based on the data from only one hospitalization record and do not include historical records even if they are available. However, from this study we can conclude that it is advantageous to generate separate models for predicting readmissions on more complex patients including the data from their historical hospitalizations, since they form a more homogenous dataset and consequently present a more complex classification problem.

### Funding
This research was supported by the Swiss National Science Foundation through a SCOPES 2013 Joint Research Projects grant SNSF IZ73Z0_152415. The funders had no role in study design, data collection and analysis, decision to publish, or preparation of the manuscript.

### Grant Disclosures
The following grant information was disclosed by the authors:
Swiss National Science Foundation.
SNSF: IZ73Z0_152415.

### Competing Interests
Zoran Obradovic is an Academic Editor for PeerJ.

### Author Contributions
- Petra Povalej Brzan conceived and designed the experiments, analyzed the data, wrote the paper, prepared figures and/or tables, reviewed drafts of the paper.

- Zoran Obradovic conceived and designed the experiments, wrote the paper, reviewed drafts of the paper.
- Gregor Stiglic conceived and designed the experiments, analyzed the data, wrote the paper, reviewed drafts of the paper.

## Data Availability

The data is available at the HCUP: http://www.hcup-us.ahrq.gov/sidoverview.jsp.

## Supplemental Information

Supplemental information for this article can be found online at http://dx.doi.org/10.7717/peerj.3230#supplemental-information.

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
