# Peer review of "Contribution of temporal data to predictive performance in 30-day readmission of morbidly obese patients"

_PeerJ, doi:10.7717/peerj.3230_

## Round 0.1 · original submission · Major Revisions

· Academic Editor

Major Revisions

Please address carefully the points raised by the two reviewers.

Reviewer 1 ·

Basic reporting

The manuscript is well-written and there are certainly no problems with the choice of wordings. However, the manuscript is not well structured for the reader.

For example, there is no need to specify a 30-day cut-off in the first sentence of the abstract to be mentioned again in the following sentence that describes current research practices. The objective sentence for the purpose of the study does not convey a proper research question. The Methods section does not flow, where the sample sizes of each data are confusing. Please consider structuring the methods section, starting with the data source and the inclusion criteria to make up the five different, yet, overlapping sample populations. It is also not clear why some of the methods were used. (i.e. regularized logistic regression, independent t-tests and ANOVA). Similarly, the result section contains overlapping information, statistics based on methods not discussed in the methods section (i.e. ROC). Please consider providing more numbers in the results section rather than descriptive sentences.

Please consider moving the section “Regularized Logistic Regression” into the statistical analyses section. The authors also seem to cite an R package to fit regularized logistic regression models, but do not explicitly state the software they’ve used in their analyses. I can also see that the objective function for regularized logistic regression is a clipped screenshot (https://web.stanford.edu/~hastie/glmnet/glmnet_alpha.html). Please consider using Mathtype if you’re using Microsoft Word, or other similar programs to neatly write out the minimization problem. These equations should be centered as well.

Overall, I would recommend to 1) improve the organization and flow of information in the manuscript, and 2) avoid or simplify run-on sentences that are not direct to the point.

In addition, please consider adjusting the manuscript figures by using different line symbols for each of the five data sets.

Experimental design

This study uses an admission database. Please check the ICD-9 code for morbid obesity. It’s 278.01, rather than 27.801 as stated in the manuscript (http://icd9.chrisendres.com/index.php?action=search&srchtext=278.01). Please note that ICD-9 codes in admission datasets are prone to false reporting due the influence of health insurance on medical practices. It is also not clear how the most frequent diagnosis were selected, and an arbitrary cut-off of 3% was used, but the authors do not state the pool sample that was used to obtain diagnosis frequencies. It would indicate whether different dichotomized predictors were used in each database.

Most importantly, I have major concerns with the applied statistical methodologies in this the study. It is incorrect to compare the performance of fitted regularized logistic regression models across overlapping samples using t-tests and ANOVA. It is also not clear why P-values were not reported (<0.01?). Why are t-test or ANOVA test statistics, which are asymptotic test statistics, are applied when the sampling distributions are already derived using 100 bootstrap samples. The authors also seem to not fully grasp that 10-fold cross validations are used to select the tuning parameter in regularized regression, and it seems that bootstrap replicates involved models with different turning parameters, and hence different number of predictors, that could be available or absent in different databases. It is also interesting that 100 bootstrap replications were used to estimate the variability in their results, but I am not sure if this is correct since they are not based on the same fitted model in each replication. There are also no specification on the type of bootstrap used in this study. The study seems to also include assessments on the number of selected predictors, but do not realize that larger samples (i.e D_1) permit more stable models with more predictors compared to those smaller samples (i.e. D_5).

Validity of the findings

no comment

Additional comments

This study is valuable and would help improve patient care in practice, but the current applied statistical methodologies are not appropriate in order to draw valid conclusions. I would highly recommend the authors to look into random forests. Random forests do not suffer from overfitting and are robust to the presence of perfect separators to allow for the inclusion of more predictors. Random forests are fitted with X number of fitted classification trees, each having Y number of terminal nodes, and R number of bootstrap samples. Please see examples in the book, The Elements of Statistical Learning - Second Edition, and the randomForest R package as well

·

Basic reporting

This article is well written; the background is clearly exposed.
The structure of the article follows the classical IMRAD structure.
The figures are correctly designed and clear.

Experimental design

Whereas a reduction of readmissions of morbidly obese patients in the 30-days after discharge represents an important challenge for many hospitals, cross-sectional predictive models have been built to predict the occurrence of readmission within 30-days, mainly based on information from the current hospitalization. This research aim to demonstrate the gain of a predictive performance obtained by inclusion of information from historical hospitalization records. This question is clear and well defined.

In terms of method, the authors used the California Statewide inpatient database to test their hypothesis (n=18,881), with an extraction from historical patient hospitalization records in a one year timeframe.
- The authors have to mention the ethical approval for using these data.
- A better description of the database (exhaustiveness of the recording, quality of the disease encoding, criteria to define/encode morbid obesity) could be added in the method (adding references using these database could also support that).

Statistical models and analysis have been rigorously conducted.

Validity of the findings

The authors have demonstrated a gain of predictive performance when including information from up to three historical records, but not with more than three historical records.
- A model without inclusion of historical records could be added in the comparisons (as reference), to better estimate the gain of historical record using, compared to a model not using historical records; this would reinforce their hypothesis.

The discussion is adapted and linked to the original research question.
- The applicability of their predictive model in the healthcare system could be also discussed, as well as its potential consequences (prevention of readmissions, better prediction of healthcare cost).

The conclusion is well stated and clear.

Additional comments

no more general comments

---

## Round 0.2 · accepted · Accept

· Academic Editor

Accept

The authors have addressed the questions of the two reviewers, thank you.